# Algebraic Morphology of DNA–RNA Transcription and Regulation

Michel Planat [1,*,†] , Marcelo M. Amaral [2,†] and Klee Irwin [2,†]

1    CNRS, Institut FEMTO-ST, Université de Franche-Comté, F-25044 Besançon, France
2    Quantum Gravity Research, Los Angeles, CA 90290, USA
*    Correspondence: michel.planat@femto-st.fr
†    These authors contributed equally to this work.

**Abstract:** Transcription factors (TFs) and microRNAs (miRNAs) are co-actors in genome-scale decoding and regulatory networks, often targeting common genes. To discover the symmetries and invariants of the transcription and regulation at the scale of the genome, in this paper, we introduce tools of infinite group theory and of algebraic geometry to describe both TFs and miRNAs. In TFs, the generator of the group is a DNA-binding domain while, in miRNAs, the generator is the seed of the sequence. For such a generated (infinite) group $\pi$, we compute the $SL(2, \mathbb{C})$ character variety, where $SL(2, \mathbb{C})$ is simultaneously a 'space-time' (a Lorentz group) and a 'quantum' (a spin) group. A noteworthy result of our approach is to recognize that optimal regulation occurs when $\pi$ looks similar to a free group $F_r$ ($r = 1$ to $3$) in the cardinality sequence of its subgroups, a result obtained in our previous papers. A non-free group structure features a potential disease. A second noteworthy result is about the structure of the Groebner basis $\mathcal{G}$ of the variety. A surface with simple singularities (such as the well known Cayley cubic) within $\mathcal{G}$ is a signature of a potential disease even when $\pi$ looks similar to a free group $F_r$ in its structure of subgroups. Our methods apply to groups with a generating sequence made of two to four distinct DNA/RNA bases in $\{A, T/U, G, C\}$. We produce a few tables of human TFs and miRNAs showing that a disease may occur when either $\pi$ is away from a free group or $\mathcal{G}$ contains surfaces with isolated singularities.

**Keywords:** transcription factors; microRNAs; diseases; finitely generated group; $SL(2, \mathbb{C})$ character variety; algebraic surfaces

## 1. Introduction

> Again since the one face, constant in symmetry, appears sometimes fair and sometimes not, can we doubt that beauty is something more than symmetry, that symmetry itself owes its beauty to a remoter principle?
>
> [1] (Ennead I, Sixt Tractate, p66).

The remote principle envisaged by Plotinus is still a symmetry principle but in a modern definition involving group theory and algebraic geometry. Recently, we wrote a paper about a common algebra possibly ruling the beauty and structure in poems, music and proteins [2]. We found that free groups govern the structure of such disparate topics where a language emerges from pure randomness. We coined the concept of 'syntactical freedom' for qualifying the occurrence of symbols and rules organized according to group theory and aperiodicity [3,4]. One favorite decomposition of the secondary structure of proteins is in term of $\alpha$-helices, $\beta$-sheets and coils [5] but this decomposition and the resulting syntax is model dependent [2]. According to our view, at the genome scale, the escape to 'syntactical freedom' means a lack of harmony and the signature of a potential disease. The secondary structures in the sequence of proteins or viruses are, most of the time, organized according to the rules of free groups and, otherwise they may be a witness of a potential aberrant topology.

Of course, there are other techniques to identify a potential disease in the DNA/RNA structures. The loose of homochirality of DNA may indicate an age-related disease [6]. Data mining techniques may be employed to identify cancer in gene expression [7]. The role of miRNAs in the regulation of pathogenesis during infection is investigated in [8].

Apart from the canonical double helix B-DNA we now know that there exists a diversity of non-canonical coding/decoding sequences organized in structures such as Z-DNA (often encountered in transcription factors [9,10]), G-quadruplex (in telomeres) and other types that are single-stranded, two-stranded, or multistranded [11]. RNA is usually a single-stranded molecule in a short chain of nucleotides, as is the case for a messenger RNA (mRNA) or a (non-coding) microRNA (miRNA).

In our approach, the investigated sequences define a finitely generated group $f_p$ whose structure of subgroups is close or away from a free group $F_r$ of rank $r$, where $r + 1$ is the number of distinct amino acids in the sequence (or the number of distinct secondary structures considered in the protein chain) [4]. Recently, we also introduced concepts for representing the groups $f_p$ over the Lie group $SL(2, \mathbb{C})$. The $SL(2, \mathbb{C})$ character variety of $f_p$ and its Groebner basis are topological ingredients, they feature algebraic geometric properties of the group $f_p$ under question [12–14]. A reminder of the theory is given in the next section.

In this paper, we focus upon transcription factors and miRNAs, both serve at properly decoding and regulating the genes and their action, either independently of each other or together by targeting common genes [15]. Figure 1 (Left) is a picture of the pluripotent transcription factor Nanog. Figure 1 (Right) is an example of a pre-miRNA associated to a disease [16]. Both are investigated in detail in this paper.

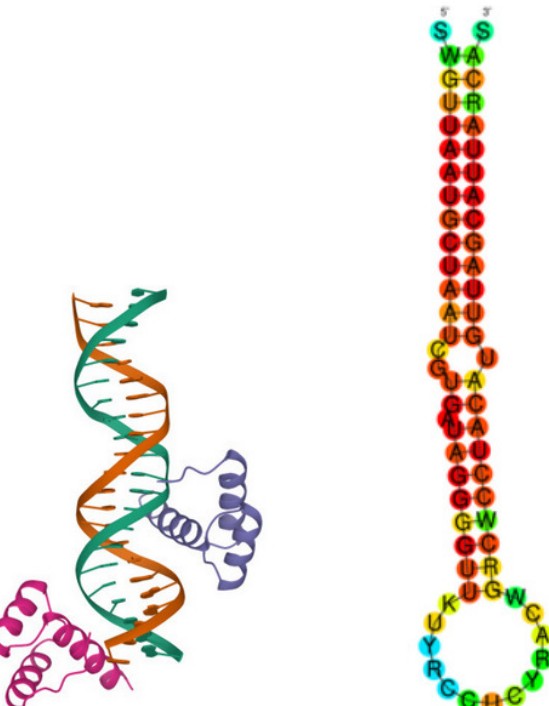

**Figure 1. Left**: the Nanog transcription factor (PDB 9ANT). **Right**: the pre-miR-155 secondary structure [16].

In Section 2, we describe the mathematical methods and the software needed for describing the algebraic surfaces relevant to DNA/RNA sequences. This includes the definition of infinite groups under question, of the free groups $F_r$ of rank $r = 1$ to 3 corresponding to 2- to 4-base sequences and the calculation of $SL(2, \mathbb{C})$ representation of such groups. A special care is needed to compute a Groebner basis of the character variety.

Section 3 is a discussion about the type of singular surfaces encountered in this research. They play an important role in our view of the discrimination of a potential disease.

In Section 4, the methods are applied to representative examples of sequences taken from transcription factors and microRNAs whose group is close or away from a free group, and whose Groebner basis of the variety contains simple singularities. Our theory applies to many TFs and miRNAs that are known to be related to an identified disease.

Section 5 summarizes our paper and opens a few perspectives.

## 2. Theory

### 2.1. Finitely Generated Groups, Free Groups and Their Conjugacy Classes

Let $F_r$ be the free group on $r$ generators ($r$ is called the rank).

The number of conjugacy classes of $F_r$ of a given index $d$ is known and is a signature of the isomorphism, or the closeness, of a group $\pi$ to $F_r$ [4,17]. The cardinality structure of conjugacy classes of index $d$ in $F_r$ will be called the card seq of $F_r$. We need the cases from $r = 1$ to 3 that correspond to the number of distinct bases in a DNA/RNA sequence. The card seq of $F_r$ is in Table 1 for the 3 sequences of interest in the context of DNA/RNA.

The free group $F_1$ of rank 1 may be defined as $F_1 = \langle a | \varnothing \rangle$ (with one generator $a$ and no relation) or $F_1 = \langle a, b | ab \rangle$ (with two generators $a$ and $b$ and the relation $ab$). Similarly, the free group $F_2$ of rank 2 is $F_2 = \langle a, b | \varnothing \rangle = \langle a, b, c | abc \rangle$ and so on for higher rank $r$.

**Table 1.** The counting of conjugacy classes of subgroups of index $d$ in the free group $F_r$ of rank $r = 1$ to 3. The last column is the index of the sequence in the on-line encyclopedia of integer sequences [18].

| $r$ | Card Seq | Sequence Code |
|---|---|---|
| 1 | $[1, 1, 1, 1, 1, 1, 1, 1, 1, \cdots]$ | A000012 |
| 2 | $[1, 3, 7, 26, 97, 624, 4163, 34470, 314493, \cdots]$ | A057005 |
| 3 | $[1, 7, 41, 604, 13753, 504243, 24824785, 1598346352, \cdots]$ | A057006 |

Next, given a finitely generated group $fp$ with a relation (rel) given by a sequence motif, we are interested in the number of conjugacy classes of subgroups of index $d$ (the card seq of $fp$). Often, for a selected DNA/RNA motif taken as the generator of a finitely generated group $fp$, the card seq that is obtained is close to that of a free group $F_r$, with $r + 1$ being the number of distinct bases involved in the motif.

The closeness of $fp$ to $F_r$ can be checked by its signature in the finite range of indices of the card seq.

To illustrate our methodology, let us consider the seed UCCUACA of the microRNA sequence mir-155-3p that is investigated in detail in Section 4.3; see also Figure 2. The finitely generated group is $fp = \langle U, A, C | UCCUACA \rangle$. The card seq of $fp$ is found to be the series $[1, 3, 10, 51, 164, 1230 \cdots]$. The group $fp$ is of rank 2 and the card seq corresponds to that of the group we call $\pi_2$ in the tables of Section 4. In this case, the card seq of $fp$ is 'away' from the card seq of the free group of the same rank $F_2$ and the two groups $fp$ and $F_2$ are of course 'away' to each other. However, if we omit the last nucleotide A of the seed in the generator of $fp$, then the group $fp$ obtains the same card seq than $F_2$ (at least in the finite range of the card seq series that we can check) and, in this sense, both groups $fp$ and $F_2$ become 'close' to each other.

### 2.2. The $SL(2, \mathbb{C})$ Character Variety of a Finitely Generated Group and a Groebner Basis

Let $fp$ be a finitely generated group, we describe the representations of $fp$ in $SL(2, \mathbb{C})$, the group of ($2 \times 2$) matrices with complex entries and determinant 1. The group $SL(2, \mathbb{C})$ may be seen simultaneously as a 'space-time' (a Lorentz group) and a 'quantum' (a spin) group.

Such a group describes representations as degrees of freedom for all quantum fields and is the gauge group for Einstein–Cartan theory. The later contains the Einstein–Hilbert action and Einstein's field equations [19]. The so-called Holst action used in loop quantum gravity has quantum gravity states given in terms of $SL(2, \mathbb{C})$ representations [20].

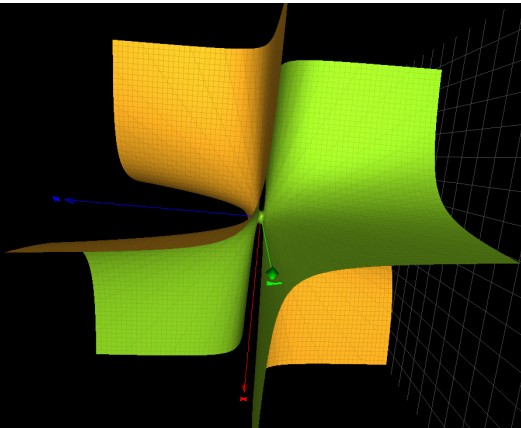

**Figure 2.** (**Up**): Complementary base-pairing between miR-155-3p and the human Irak3 (interleukin-1 receptor-associated kinase 3) mRNA ([16], Figure 5). The requisite 'seed sequence' base-pairing is denoted by the bold dashes. (**Down**): the surface $f_b^{(A_1)}(x,y,z) = x^2 + y^2 - 6z^2 + 4xyz$.

Representations of $f_p$ in $SL(2,\mathbb{C})$ are homomorphisms $\rho : f_p \to SL(2,\mathbb{C})$ with character $\kappa_\rho(g) = \mathrm{tr}(\rho(g))$, $g \in f_p$. The notation $\mathrm{tr}(\rho(g))$ means the trace of the matrix $\rho(g)$. The set of characters allows to determine an algebraic set by taking the quotient of the set of representations $\rho$ by the group $SL(2,\mathbb{C})$, that acts by conjugation on representations [21,22].

Such an algebraic set is called the $SL(2,\mathbb{C})$ character variety of $f_p$. The set is made of a sequence of multivariate polynomials called a scheme $X$. The ideal $\mathcal{I}(X)$ is defined by the vanishing of polynomials living in the scheme $X$. Below, we use a particular basis $\mathcal{G}(X)$ of the polynomial ring $\mathcal{I}(X)$, called a Groebner basis. The Groebner basis $\mathcal{G}(X)$ has to follow algorithmic rules (similar to the Euclidean division for univariate polynomials) [14].

For the effective calculations of the character variety, we make use of a software on Sage [23]. We also need Magma [24] for the calculation of a Groebner basis, at least for 3- and 4-base sequences.

### 2.3. Algebraic Geometry and Topology of DNA/RNA Sequences

2.3.1. Two-Base Sequences

Following [25], in this section, we are interested by the special case of representations for the once punctured torus $S_{1,1}$ and the relevance of the extended mapping class group $\mathrm{Mod}^{\pm}(S_{1,1})$ in its action on surfaces of type $\kappa_d(x,y,z)$, $d \in \mathbb{C}$.

The fundamental group of $T_{1,1}$, that we denote $\pi$, is the free group $F_2 = \langle a,b|\varnothing \rangle$ on two generators $a$ and $b$. The boundary component of $T_{1,1}$ consists of a single loop around the puncture expressed by the commutator $[a,b] = abAB$ with $A = a^{-1}$ and $B = b^{-1}$. Taking the traces $x = \mathrm{tr}(\rho(a))$, $y = \mathrm{tr}(\rho(b))$, $z = \mathrm{tr}(\rho(ab))$, the trace of the commutator is the surface [21,25]

$$\mathrm{tr}([a,b]) = \kappa_2(x,y,z) = x^2 + y^2 + z^2 - xyz - 2.$$

According to the Dehn–Nielsen–Baer theorem [26], for a surface of genus $g \geq 1$, we have

$$\mathrm{Mod}^{\pm}(S_g) \cong \mathrm{Out}(\pi(S_g)),$$

where $\mathrm{Mod}(S)$ is the mapping class group. It denotes the group of isotopy classes of orientation-preserving diffeomorphisms of $S$.

The extended mapping class group $\text{Mod}^\pm(S)$ is the group of isotopy classes of all homeomorphisms of $S$ (including the orientation-reversing ones). The outer automorphism group of $\pi$ is denoted $\text{Out}(\pi)$. This leads to the (topological) action of $\text{Mod}^\pm$ on the once punctured torus as follows

$$\text{Mod}^\pm(S_{1,1}) = \text{Out}(F_2) = GL(2,\mathbb{Z}). \tag{1}$$

The automorphism group $\text{Aut}(F_2)$ acts by composition on the representations $\rho$ and induces an action of the extended mapping class group $\text{Mod}^\pm$ on the character variety by polynomial diffeomorphisms of the surface $\kappa_d$ defined by [25]

$$f_H^{(4)}(x,y,z) = \kappa_d(x,y,z) = xyz - x^2 - y^2 - z^2 + d. \tag{2}$$

Within the family $\kappa_d(x,y,z)$, the Cayley surface $\kappa_4(x,y,z)$ is obtained from the character variety for the fundamental group of the Hopf link $L2a1$ (the link of two unknotted curves). For the Hopf link, the fundamental group is

$$\pi(S^3 \setminus L2a1) = \langle a,b|[a,b]\rangle = \mathbb{Z}^2,$$

where the notation $S^3$ is for the 3-sphere and $\setminus L2a1$ means that a small tubular neighborhood around the Hopf link $L2a1$ is removed from $S^3$ to define the corresponding Hopf link 3-manifold.

The Cayley surface $\kappa_4(x,y,z)$ possesses 4 simple (isolated) singularities. At these points, the surface loses its smoothness. We already showed that it plays a role in the context of Z-DNA conformations of transcription factors ([13], Tables 2 and 5); see also, Section 4 below, and notably Figure 3 (Left).

The surface $\kappa_3(x,y,z)$ lies within the character variety for the fundamental group of the link L6a1 [27]. We show below that this surface also lies in the generic Groebner basis obtained for 4-base sequences; see Section 4 below.

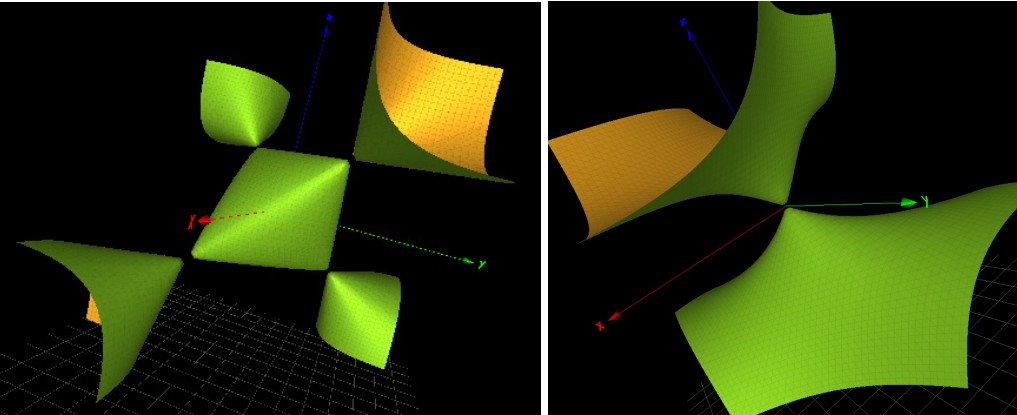

**Figure 3.** (**Left**): the Cayley cubic $\kappa_4(x,y,z)$. (**Right**): the surface $f_a^{(A_1)}(x,y,z)$.

2.3.2. Three-Base Sequences

Our main object in this section is the four punctured sphere for which the fundamental group is the free group $F_3$ of rank 3 whose character variety generalizes the Fricke cubic surface (2) to the surface $V_{a,b,c,d}(\mathbb{C})$ in $\mathbb{C}^7$. From now, $a$ and $b$ are no longer generators of a group but belong to the set of parameters of the surface $V$.

We follow the work of references [21,25,28].

Let $S_{4,2}$ be the quadruply punctured sphere. The fundamental group for $S_{4,2}$ can be expressed in terms of the boundary components $x_1$, $x_2$, $x_3$, $x_4$ as
$\pi(S_{4,2}) = \langle x_1, x_2, x_3, x_4 | x_1 x_2 x_3 x_4 \rangle \cong F_3$.

A representation $\pi \to SL(2, \mathbb{C})$ is a quadruple

$$\alpha = \rho(A), \ \beta = \rho(B) \ \gamma = \rho(C), \ \delta = \rho(D) \in SL(2, \mathbb{C}) \ \text{where} \ \alpha\beta\gamma\delta = I.$$

Let us associate the seven traces

$$a = \text{tr}(\rho(\alpha)), \ b = \text{tr}(\rho(\beta)), \ c = \text{tr}(\rho(\gamma)), \ d = \text{tr}(\rho(\delta))$$
$$x = \text{tr}(\rho(\alpha\beta)), \ y = \text{tr}(\rho(\beta\gamma)), \ z = \text{tr}(\rho(\gamma\alpha)),$$

where $a, b, c, d$ are boundary traces and $x, y$ and $z$ are traces of elements $AB$, $BC$ and $CA$ representing simple loops on $S_{4,2}$.

The character variety for $S_{4,2}$ satisfies the equation ([21], Section 5.2), ([25], Section 2.1), ([28], Section 3B), ([29], Equation (1.9)) or ([30], Equation (39)).

$$V_{a,b,c,d}(x, y, z) = x^2 + y^2 + z^2 + xyz - \theta_1 x - \theta_2 y - \theta_3 z - \theta_4 = 0 \tag{3}$$

with $\theta_1 = ab + cd$, $\theta_2 = ad + bc$, $\theta_3 = ac + bd$ and $\theta_4 = 4 - a^2 - b^2 - c^2 - d^2 - abcd$.

The 4-punctured sphere, whose fundamental group is the free group $F_3$ with generator the product of the 4 letters, is a generic topology. It is straightforward to check that the Groebner basis for $F_3$ contains (among other surfaces and depending on the choice of parameters) a single copy of the generic surfaces $\kappa_4(x, y, z)$, $\kappa_3(x, y, z)$ and $V_{1,1,1,1}(x, y, z) = xyz + x^2 + y^2 + z^2 - 2x - 2y - 2z + 1$, a surface we also denote $f^{(3A_1)}(x, y, z)$ because it contains 3 simple singularities of type $A_1$ as shown in Figures 4 and 5.

There are other surfaces encountered in our study of the Groebner basis for transcription factors and miRNAs when the generated group is close or away from the free group $F_2$ (for 3-base sequences) or the free group $F_3$ (for 4 base sequences). These surfaces are described in Section 4.

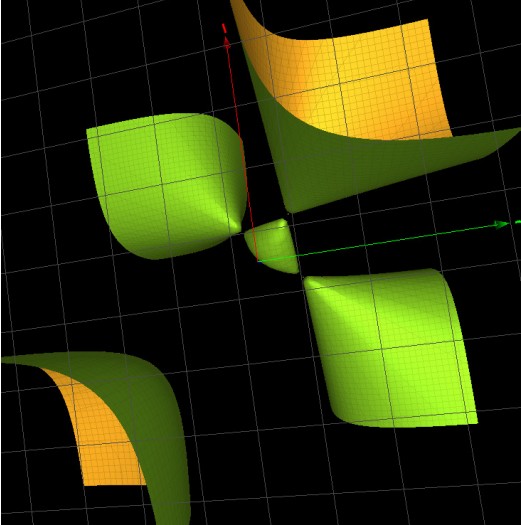

**Figure 4.** The Fricke surface $V_{1,1,1,1}(x, y, z) = f_a^{(3A_1)}(x, y, z)$ (with three simple singularities of type $A_1$).

### 2.3.3. Four-Base Sequences

There does not exist a huge difference in the structure of a Groebner basis of the character variety in the case of a 4-base sequence compared to the case of a 3-basis sequence. One difference is that one has to manage a 14-dimensional hypersurface $V_{a,b,c,d,e,f,g,h}(x, y, z, u, v, w)$ in $\mathbb{C}^{14}$ (instead of a 7-dimensional one as in the previous subsection). In general, after the appropriate choice of the 8 parameters $a, b, c, d, e, f, g, h$, the Groebner basis contains more than one copy of the generic Groebner basis. Each copy $S$ of a relevant surface may be of the form $S(x, y, z)$, $S(x, u, v)$, $S(y, u, w)$ or $S(z, v, w)$.

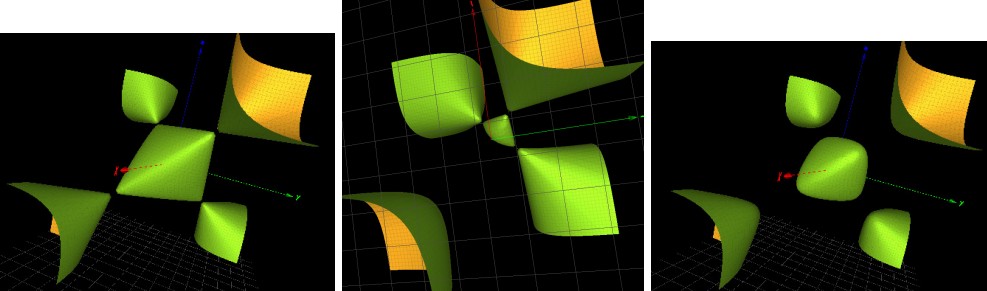

**Figure 5.** (**Up**): Complementary base-pairing between miR-155-5p and the human Spi1 (spleen focus forming virus proviral integration oncogene) ([16], Figure 4). The requisite 'seed sequence' base-pairing is denoted by the bold dashes. (**Down (from left to right)**): the surfaces $f_H^{(4)} = \kappa_4(x,y,z)$, $f^{(3A_1)}(x,y,z)$ and $\kappa_3(x,y,z)$, four copies of them are contained within the Groebner basis for the character variety.

## 3. Discussion

Given an ordinary projective surface $S$ in the projective space $P^3$ over a number field, when $S$ is birationally equivalent to a rational surface, the software Magma [24] determines the map to such a rational surface and returns its type within five categories. The returned type of $S$ is $P^2$ for the projective plane, a quadric surface (for a degree 2 surface in $P^3$), a rational ruled surface, a conic bundle or a degree $p$ Del Pezzo surface where $1 \leq p \leq 9$.

One important attribute of a projective surface is its degree of singularity. Most surfaces $S$ of interest below are almost not singular in the sense that they have at worst simple (isolated) singularities. A simple singularity is referred to as an A-D-E singularity [31]. It has to be of the type $A_n$, $n \geq 1$, $D_n$, $n \geq 4$, $E_6$, $E_7$ or $E_8$.

The type and the number of simple singularities are denoted in an exponent such as $S^{(lA_1)}$ for l singularities of type $A_1$. All such surfaces are degree 3 del Pezzo surfaces. For the Cayley cubic $f^{(4)}(x,y,z)$, the exponent (4) means $(4A_1)$.

There are additional facilities offered by Magma for studying a singular scheme. As already mentioned, a scheme is any geometric object defined by a set of polynomials in a projective space. An algebraic surface is a scheme. If the scheme has simple singularities, one can calculate the degree and the support of the reduced singular subscheme that are signatures of the scheme. In the examples of this paper, for a surface $S^{(lA1)}$, the degree is $l$ and the support contains $l$ simple singular points. Otherwise, we add a lower index to $S^{(lA1)}$ to qualify the index and the support of the singular subscheme. The notation $S_{m,\{\}}^{(lA1)}$ means that there are $l$ singularities of type $A_1$, that the degree of the singular subscheme is $m$ and that the support is the empty set.

Most of the time, the surfaces in the character variety attached to a transcription factor or a microRNA only contain isolated singularities.

This contrasts with some sequences encountered in another context. For instance, a complete turn of A-DNA (PDB; 2D47) defines the dodecamer sequence CCCCCGCGGGGG whose attached character variety contains the surface ([13], Figure 5).

$$f_{\tilde{H}}(x,y,z) = z^4 - 2xyz + 2x^2 + 2y^2 - 3z^2 - 4. \tag{4}$$

This surface contains many non isolated singularities (that cannot be resolved by blow ups). Methods to desingularize a surface of such a type are presented in a companion paper [32].

To summarize the important issues below, a noteworthy result of our approach is to recognize that optimal regulation occurs when the group underlying the sequence looks similar to a free group $F_r$ ($r = 1$ to 3) in the cardinality sequence of its subgroups, a result obtained in our previous papers. A non-free group structure features a potential disease. A second noteworthy result is about the structure of the Groebner basis of the variety. A surface with simple singularities (such as the well known Cayley cubic) within the Groebner basis is a signature of a potential disease even when the generated group looks similar to a free group $F_r$ in its structure of subgroups. Our methods apply to groups with a generating sequence made of two to four distinct DNA/RNA bases in $\{A, T/U, G, C\}$. Several human TFs and miRNAs are investigated in detail thanks to our approach. We summarize this discussion in the diagram presented in Figure 6.

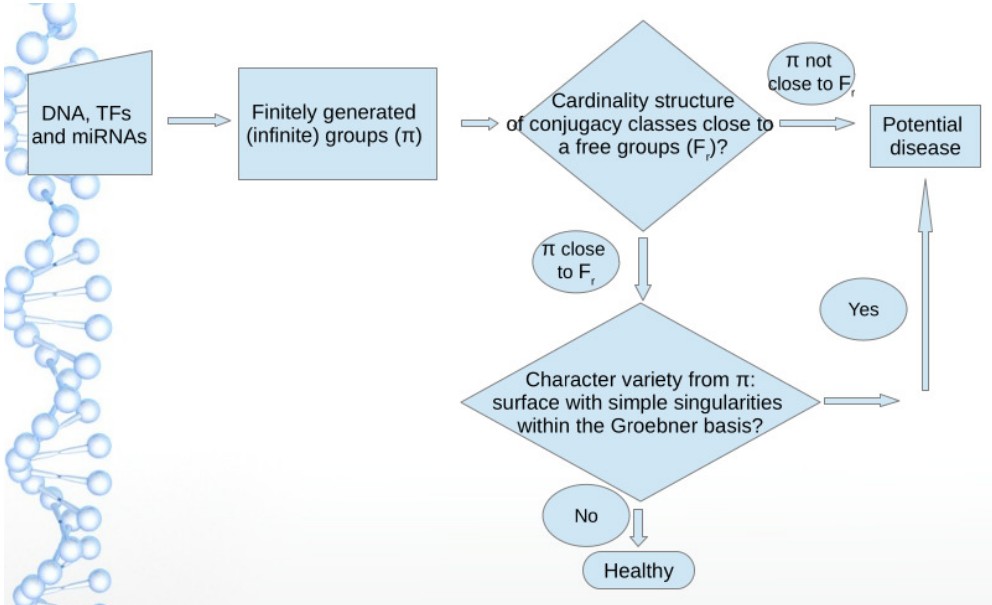

**Figure 6.** A diagram with the main results discussed in the main text.

## 4. Results

In this section, we apply the $SL(2, \mathbb{C})$ representation theory to groups generated by DNA/RNA sequences occurring in transcription factors and microRNAs. Both play a leading role in the decoding of the genome and in genome-scale regulatory networks. Two-letter transcription factors (TFs) whose structure is close or away from the free group $F_1$ were already investigated in ([13], Table 2). The occurrence of the Cayley cubic $\kappa_4(x, y, z)$ in the Groebner basis of the character variety was found to be a signature in the former case. In this case, this surface seems to possess a regulatory action that may be lost in the latter case. In ([13], Table 3), the potential diseases associated with a non-free group structure are mentioned. In the present section, one explores, in Table 2, the case of a 3-letter sequence of a TF and in Table 3, the case of 3-letter sequence of a miRNA. The case of a 4-letter sequence of a miRNA is summarized in Table 4. The role of surfaces with simple singularities in the Groebner basis is emphasized.

### 4.1. Algebraic Morphology of the Transcription Factor Prdm1

The transcriptional repressor PR domain containing 1 (Prdm1), also known as B-lymphocyte-induced maturation protein-1 (Blimp1), is essential for normal development and immunity [33]. It is of a zinc finger type. The consensus sequence ACTTTC corresponds to the code MA0508.2 in [34].

#### 4.1.1. The Character Variety

The ideal for the character variety $f_{\mathrm{Prdm1}}(a, b, c, d)(x, y, z)$ for a few values of the parameters is

$$f_{\mathrm{Prdm1}}(0, 0, 0, 0) = \kappa_{-4}(x, y, z)(yz + x + 2),$$

$$f_{\mathrm{Prdm1}}(0, 1, 1, 0) = y\kappa_{-2}(x, y, z)(x - 1),$$

$$f_{\mathrm{Prdm1}}(0, 1, 0, 0) = z\kappa_{-3}(x, y, z)(z^2 + 1)(yz + x + 1)(yz + x + 2),$$

$$f_{\mathrm{Prdm1}}(1, 1, 1, 1) = f_a^{(3A_1)}(x, y, z)(y + 1)(y + z - 1),$$

where $\kappa_{-2}(x, y, z)$, $\kappa_{-3}(x, y, z)$ are Fricke surfaces [27] and $f_a^{(3A_1)}(x, y, z) = xyz + x^2 + y^2 + z^2 - 2x - 2y - 2z + 1$ is the surface drawn in Figure 4. The subscript $3A_1$ is for featuring the three singularities of type $A_1$.

#### 4.1.2. The Groebner Basis

The singular surfaces found in the Groebner basis of the ideal are not similar to those in the ideal. One of them $S_1 = S_{3,\{0:1:0:0\}}^{(A_1 A_3)} = 2yz^2 + x^2 + 3z^2 - 2xz - 2yz - 2y - 2z - 2$ is obtained at values $(a, b, c, d) = (1, 1, 1, 1)$ featuring two simple singularities of type $A_1$ and $A_3$ with a singular subscheme of degree 3 and the singular point of type $A_3$ in its support. The other surface obtained in the Groebner basis at values $(a, b, c, d) = (0, 0, 0, 0)$ is a conic bundle of the $K_3$ type $S_2 := z^4 + 2yz^3 + x^2 - 6yz - 2x - 8$ whose singular subscheme is non zero dimensional and of degree 1.

These two surfaces are non standard in our context of TFs and miRNAs. The formal desingularization of the surface $S_2$ is given in [32].

### 4.2. Algebraic Morphology of Homeodomains for Nanog and Xvent

The pluripotency in embryonic stem cells and their regulation is characterized by the expression of several transcription factors [35,36]. Among them, the transcription factor Nanog is present in the embryonic stages of life of several vertebrate species. Nanog binds to promoter elements of hundreds of target genes as a regulatory element. It has a conserved DNA-binding homeodomain with consensus sequence TAATGG. The closest homolog of Nanog is the (nonmammalia) Xenopus, a Xvent transcription factor with consensus sequence CTAATT [36]. In this subsection, we investigate the algebraic morphology of both transcription factors Nanog and Xvent thanks to their consensus sequences.

**Table 2.** A few (three-base) transcription factors whose group structure is away from a free group or whose Groebner basis of the $SL(2, \mathbb{C})$ character variety contains a (possibly almost) singular surface. The symbol gene is for the identification of the transcription factor in the Jaspar database [34], motif is for the consensus sequence of the transcription factor, card seq is for the cardinality sequence of conjugacy classes of subgroups of the group whose motif is the generator, simple sing is for the identification of a surface with simple singularities within the Groebner basis and the last column is for a reference paper and the corresponding disease. The group $F_2$ is the free group of rank two. The card seq for $\pi_2$ is $[1, 3, 10, 51, 164, 1230, 7829, 59835, 491145 \cdots]$, close to the card seq of the group $\langle x, y, z | (x, (y, z)) = z \rangle$. The latter group is found as governing the structure of many transcription factors and is associated to the link found in ([13], Figure 2). The card seq for $\pi_3$ is $[7, 14, 89, 264, 1987, 11086, 93086 \cdots]$. The surface $f_b^{(A_1)}(x, y, z) = x^2 + y^2 - 6z^2 + 4xyz$ (not defined in the text) is part of the character variety for the genes Pitx1, OTX1, etc.

| Gene | Motif | Card Seq | Simple Sing | Ref & Disease |
|---|---|---|---|---|
| Prdm1 | ACTTTC | $F_2$ | $S_1, S_2(x, y, z)$ | [34], MA0508.2 lupus, rheumatoid arthritis |

**Table 2.** *Cont.*

| Gene | Motif | Card Seq | Simple Sing | Ref & Disease |
|------|-------|----------|-------------|---------------|
| POU6F1 | TAATGAG | $\pi_2$ | no | MA1549.1 |
|  |  |  |  | lung adenocarcinoma |
| ELK4 | CTTCCGG | . | no , Fricke | MA0076.2 |
|  |  |  |  | gastric cancer |
| OTX2 | GGATTA | $\pi_3$ | no | [MA0712.2, MA0883.1] |
|  |  |  |  | medulloblastomas |
| N-box | TTCCGG | . | no, Fricke | [37] |
|  |  |  |  | drug sensitivity |
| Pitx1,OTX1,$\cdots$ | TAATCC | . | $f_H^{(4)}, f_b^{(A_1)}(x,y,z)$ | [34], [MA0682.1,MA0711.1] |
|  |  |  |  | autism, epilepsy, $\cdots$ |
| Nanog | TAATGG | . | $f_H^{(4)}, f_a^{(A_1)}(x,y,z)$ | [35] |
|  |  |  |  | cancer cells |
| Xvent | CTAATT | F2 | $f_{4,\{\}}^{(2A_1)}, f^{(A_2)}(x,y,z)$ | [36] |

The Groebner basis for Xvent $f_{\text{Nanog}}(0,0,0,0)$ takes the form

$$f_{\text{Nanog}}(0,0,0,0) = f_H^{(4)}(x,y,z)f_a^{(A_1)}(x,y,z)\cdots$$

where $f_H^{(4)}(x,y,z)$ is the Cayley cubic (with its 4 simple singularities) and $f_a^{(A_1)}(x,y,z) = x^2 + y^2 - z^2 + xyz$ (a surface with a single simple singularity of type $A_1$) as shown in Figure 3 (Right). The forgotten factors are factors for planes or trivial smooth surfaces.

The Groebner basis for Xvent $f_{\text{Xvent}}(1,1,1,1)$ takes the form

$$f_{\text{Xvent}}(1,1,1,1) = f_b^{(3A_1)}(x,y,z)\cdots$$

where $f_b^{(3A_1)}(x,y,z) = x^2 + y^2 + xyz - xy - z - 1$ (a surface with three simple singularity of type $A_1$). The missing term does not contain surfaces with singularities. The character variety $f_{\text{Xvent}}(0,0,0,0)$ contains the cubic surface $f_{4,\{\}}^{(2A_1)}(x,y,z) = 2z^3 + x^2z + 2xyz + 2y^2 - z^2 - 6z$ (with two simple singularities of type $A_1$) and other factors for planes or trivial smooth surfaces. Both surfaces $f_b^{(3A_1)}(x,y,z)$ and $f_{4,\{\}}^{(2A_1)}(x,y,z)$ are pictured in Figure 7.

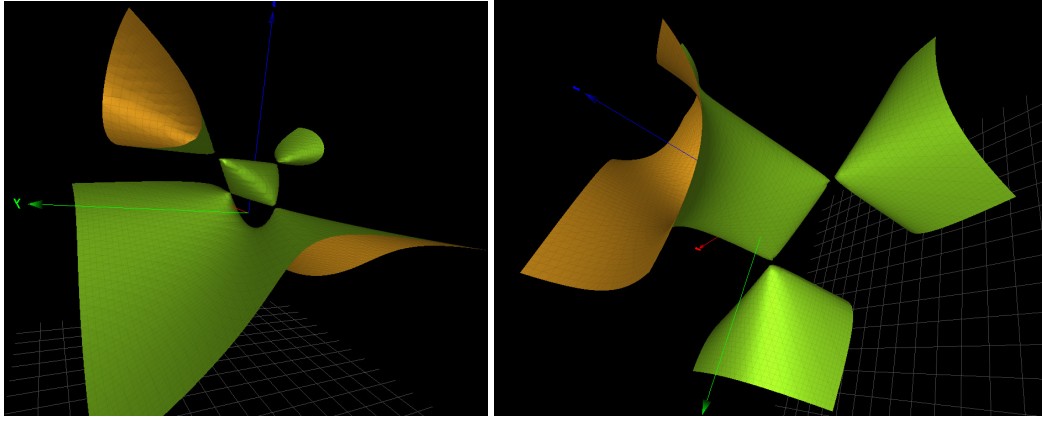

**Figure 7.** (**Left**): the cubic surface $f_{4,\{\}}^{(2A_1)}(x,y,z)$. (**Right**): the cubic surface $f_b^{(3A_1)}(x,y,z)$.

Table 2 lists a few selected transcription factors, their card seq and the corresponding singular surfaces, if any. As announced, in the selected transcription factors, there exists a correlation between the lack of 'syntactical freedom', or the presence of a surface with isolated singularities in the character variety, with an identified disease.

*4.3. Algebraic Morphology of microRNAs*

MicroRNAs (miRNAs) are important players of the expression and regulation of genes by targeting specific messenger RNAs (mRNAs) for degradation or translational repression [38,39]. The miRNAs are approximately 22 nt (where nt is for nucleotide) long single-stranded RNA molecules. The genes encoding miRNAs have a length much longer than the processed mature miRNA molecule. Many miRNAs reside in introns of their pre-mRNA host genes. They share their regulatory elements, primary transcript and have a similar expression profile. MicroRNAs are transcribed by RNA polymerase II as large RNA precursors called pri-miRNAs. The pre-miRNAs are approximately 70-nucleotides in length and are folded into imperfect stem-loop structures; see Figure 1 (Right) for an example.

Each miRNA is synthesized as a miRNA duplex comprising two strands (-5p and -3p). However, only one of the two strands is selectively incorporated into the RNA-induced silencing complex to act as a template for the transcript of a complementary mRNA [40,41]. For details about the mirRNA sequences, we use the Mir database [42,43].

Plant miRNAs usually have near-perfect pairing with their mRNA targets so that gene repression proceeds through cleavage of the target transcripts. In contrast, animal miRNAs are able to recognize their target mRNAs by using as few as 6 to 8 nucleotides (the seed region), which is not enough pairing for leading to cleavage of the target mRNAs. A given miRNA may have hundreds of different mRNA targets, and a given target might be regulated by multiple miRNAs.

Disregulation of miRNAs may lead to a disease such as cancer. A key microRNA known as an oncommir (involved in immunity and cancer) is mir-155.

Specifically the -3p strand is mir-155-3p. Figure 2 (top) illustrates the complementary base-pairing between miR-155-3p and the human IRAK3 (interleukin-1 receptor-associated kinase 3) mRNA ([16], Figure 4) and the relevant seed sequence UCCUAC. The card seq for this sequence is the two-letter free group $F_2$ and the Groebner basis for the corresponding character variety contains the surface $f_b^{(A_1)}(x, y, z) = x^2 + y^2 - 6z^2 + 4xyz$ that has a single simple singularity as shown in Figure 2 (down). If one retains the full seed sequence is UCCUAC(A) then the card seq passes to that of the free group $F_2$ to the group $\pi_2$ and the singular surface is lost. This is a case where the 'bandwidth' of the seed is critical in the (dis)regulation of the miRNA. These results are transcribed in Table 3.

**Table 3.** A few human (prefix 'hsa') microRNAs whose group structure is away from a free group or whose Groebner basis of the $SL(2, \mathbb{C})$ character variety contains a singular surface. The symbol mir is for the identification in the Mir database [43], seed is for the seed of the miRNA, card seq is for the cardinality sequence of conjugacy classes of subgroups of the group whose seed is the generator, sing is the identification of a singular surface within the Groebner basis and the last column is for a reference paper and the corresponding disease [40]. The card seq for $\pi_1$ and $\pi_1'$ are given in ([4], Table 5). The card seq for $\pi_2'$ is $[1, 3, 7, 34, 139, 931, 5208, 43867 \cdots]$. For hsa-mir-124-1-3p, one encounters the Fricke surface $f_{2,\{\}}^{(A_1)} = xyz + x^2 + y^2 + z^2 - 2y$ in the character variety.

| mir | Seed | Card Seq | Simple Sing | Ref & Disease |
|---|---|---|---|---|
| hsa-mir-193b-5p | GGGGUU | $\pi_1$ | no | [40,43] |
| | GGGGUUU | $\pi_1'$ | no | lung cancer |
| hsa-mir-155-3p | UCCUAC | $F_2$ | $f_b^{(A_1)}(x, y, z)$ | [40,41,43] |
| | UCCUACA | $\pi_2$ | no | multiple sclerosis |
| hsa-mir-193a-5p | GGGUCUU | $F_2$ | $f_b^{(A_1)}(x, y, z)$ | [40,43] |
| | | | | breast cancer |
| hsa-mir-223-5p | GUGUAUU | . | . | . |
| hsa-mir-133-3p | UUGGUC | $F_2$ | $f_b^{(3A_1)}(x, y, z)$ | [40,43] |
| | UUGGUCC | $\pi_2'$ | no | atrial fibrillation |

**Table 3.** *Cont.*

| mir | Seed | Card Seq | Simple Sing | Ref & Disease |
|---|---|---|---|---|
| hsa-mir-124-3p | AAGGCA | $F_2$ | $f_b^{(3A_1)}, f_{2,\{\}}^{(A_1)}$ | [43,44] |
| | AAGGCAC | . | no sing | Alzheimer's disease |

For the case of -5p strand mir-155-3p, the seed sequence UUAAUGCUA contains four distinct letters. This case is similar to generic Groebner bases obtained from four letter seeds. Depending on the choices of parameters $a, b, c, d, e, f, g, h$, the Groebner basis contains the Cayley cubic $f_H^{(4)}(x, y, z)$, the Fricke surface $\kappa_3(x, y, z)$ (that is related to the link L6a1 ([27], Figure 2)), the surface $f_a^{(3A_1)}(x, y, z)$ shown in Figure 4 and other surfaces. In this generic case, the surface is found with (at most) 4 copies where each copy is attached to a distinct puncture of the 4-punctured 4-sphere $S_{4,2}$.

In Table 3, this generic case is denoted $4 \times$ generic (or $3 \times$ generic for mir-133-5p). These results are transcribed in Table 4.

**Table 4.** The opposite strand of the microRNA considered in Table 3. The seed sequence is made of 4 distinct bases and the corresponding card seq is the free group $F_3$ of rank 3. The Groebner basis contains 4 copies of the generic collection of surfaces $\kappa_4(x, y, z)$, $f^{(3A_1)}(x, y, z)$, $\kappa_3(x, y, z)$, etc., as shown in Figure 5, except for the -5p strand of mir-133, where there are only 3 copies of the generic surfaces.

| mir | Seed | Card Seq | Sing | Ref & Disease |
|---|---|---|---|---|
| hsa-mir-193b-3p | ACUGGCC | $F_3$ | $4\times$ generic | [40,43] |
| hsa-mir-155-5p | UUAAUGCUA | . | . | [40,41,43] |
| hsa-mir-193a-3p | ACUGGCC | . | . | [40,43] |
| hsa-mir-223-3p | GUCAGUU | . | . | . |
| hsa-mir-124-5p | GUGUUCA | . | . | . |
| hsa-mir-133-5p | GCUGGUA | . | $3\times$ generic | [43,44] |

A small list of huma miRNAs is investigated in Tables 3 and 4 corresponding to 3-letter and 4-letter seeds. the prefix 'hsa' is for the human species. Similar to transcription factors in Table 2, the lack of 'syntactical freedom', or the occurrence of a singular surface in the character variety, is symptomatic of a disease.

## 5. Conclusions

We found, in this work, that a signature of a disease may be given in terms of the group structure of a DNA/RNA sequence and the related character variety representing the group. The DNA motif of a transcription factor, or the seed of a microRNA, defines the generator of a group $\pi$. As soon as $\pi$ is away from a free group $F_r$ (with $r + 1$ the number of distinct bases in the sequence) or the $SL(2, \mathbb{C})$ character variety $\mathcal{G}$ of $\pi$ contains singular surfaces with isolated singularities, a potential disease is on sight, as shown in Figure 6.

For example, for mir-155-3p, examined in much detail in Section 4.3, the seed UCCUAC serves as the generator of the group $\pi$ whose card seq is that of the free group $F_2$ and whose Groebner basis of the variety contains the surface $f_b^{(A_1)}(x, y, z)$ possessing an isolated singularity of type $A_1$. The potential disease is multiple sclerosis. Note that a longer seed (with A added at the right hand side) is not appropriate to the detection. A partial list of other potential diseases identified from the structure of a miRNA seed can be seen in Table 3. The diseases are a lung cancer associated with mir-193b-5p, a breast cancer associated with mir-194a-5p or mir-223-5p, an atrial fibrillation associated with mir-133-3p and an Alzheimer's disease associated with mir-124-3p.

One would like to be more predictive in identifying the potential disease with peculiar groups or singular surfaces. First of all, most of the time, the surfaces encountered in the context of TFs and miRNAs are degree 3 del Pezzo, in contrast to surfaces obtained from

other DNA sequences, as in Equation (4). However, the degree 3 del Pezzo family is very rich. For instance, the singular surface $f_b^{(A_1)} = x^2 + y^2 - 6z^2 + 4xyz$ (see Figure 2) is part of the character variety of TF Pitx1 (see Table 2) and of miRNAs 155-3p and 193a-5p (in Table 3). Then, the singular surface $f_{2,\{\}}^{(A_1)} = x^2 + y^2 + z^2 + xyz - 2y$ is part of the character variety of mirRNA 133-3p. Both surfaces have a simple singular point of type $A_1$ but distinct singular subschemes (see Section 3 for the notation).

An exception to the degree 3 del Pezzo rule was found in investigating the character variety for the Prdm1 transcription factor in Section 4.1. Do these features and other ones to be described later help for the diagnostic of a potential disease? There is room for much work in the future along these lines, as shown in our recent new paper [32].

In a separate direction, the work of Reference [45] about biochirality and the CPT theorem may be possibly related to the space-time-spin group $SL(2, \mathbb{C})$ that served as a prism of DNA/RNA structure in our approach. Related ideas about the concept of a time crystal are also worthwhile to be pointed out, e.g., Reference [46].

**Author Contributions:** Conceptualization, M.P.; methodology, M.P and M.M.A.; software, M.P.; validation, M.M.A. and K.I.; formal analysis, M.P.; investigation, M.M.A.; resources, K.I.; data curation, M.P.; writing—original draft preparation, M.P.; writing—review and editing, M.P. and M.M.A.; visualization, M.M.A.; supervision, M.P.; project administration, K.I.; funding acquisition, K.I. All authors have read and agreed to the published version of the manuscript.

**Funding:** This research received no external funding.

**Informed Consent Statement:** Not applicable.

**Data Availability Statement:** Data are available from the authors after a reasonable demand.

**Conflicts of Interest:** The authors declare no conflict of interest.

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
