# Peer review of "Algebraic Morphology of DNA–RNA Transcription and Regulation"

_symmetry, doi:10.3390/sym15030770_

Round 1

Reviewer 1 Report

Thank you for your effort and interesting work.

This paper introduces 2 tools of infinite group theory and of algebraic geometry to describe both Transcription Factors (TFs) and microRNAs (miRNAs).

The topic original or relevant in the field.

Compared with other published material, there is an obvious difference.

The paper focuses upon transcription factors and miRNAs, both serve at  properly decoding and regulating the genes and their action, either independently of each other or together by targeting common genes

The conclusions are consistent with the evidence and arguments presented.

I have some comments that must be considered in the modified manuscript.

-------------

1) I see both (Abstract) and (Conclusion) are descriptive. Please mention some numerical values of the main finding(s).

2) It is needed to mention the evaluating parameter(s) of your results. I notice that in tables, you mentioned methods used in some references, but, I do NOT see results for comparison. How do you judge your results are correct (or practical)?

3) Concerning the procedure, I lost myself ! Can you please draw a block diagram (or a flow chart) to illustrate the procedure, for the readers to easy follow it.

References are up-to-date.

Author Response

Thank you for the reading and recommendations for improving our paper.

1) We summarized our findings about miRNas in the second paragraph of the conclusion. For TF's the reader can easily switch to Table 2 with a similar effort.

3) You suggested to add a block diagram of the procedure. We did it in Figure 2. We think that this diagram much simplifies the understanding of our methodology. 

2) In tables, the references are for experiments, not for theoretical methods. For this reason, no comparison is possible.

Reviewer 2 Report

The article is very well written and presented. In particular I am not in favor of quoting oneself, only if it is necessary. I consider that more than the self-citation, it is the way it is written. I consider it very colloquial the way in which self-quotations are made.
In the theory section I consider that it is adequately explained and with the necessary details for its understanding. However, the example (Algebraic morphology of the transcription factor Prdm1), is not easy to follow. The conclusion should be more concrete, at some point it seems a bit more of a discussion.

Author Response

Thank you for the reading and recommendations for improving our paper.

1) We summarized our findings about miRNas in the second paragraph of the conclusion. For TF's the reader can easily switch to Table 2 with a similar effort.

3) Referee 1 suggested to add a block diagram of the procedure. We did it in Figure 2. We think that this diagram much simplifies the understanding of our methodology.